# Peer review of "Effect of Physical Training on Body Composition in Brazilian Military"

_ijerph, 2022, doi:10.3390/ijerph19031732_

Round 1

Reviewer 1 Report

As a suggestion for improvement, in case of authors want to consider it. The description of the training program should be complete. Although the authors have two references, they should provide more information about the training program, and not just limit themselves to mentioning the activity and weekly frequency.

Author Response

(R. = Reviewer; A. = Authors)

Reviewer 1

R.: As a suggestion for improvement, in case of authors want to consider it. The description of the training program should be complete. Although the authors have two references, they should provide more information about the training program, and not just limit themselves to mentioning the activity and weekly frequency.

A.:. We once again appreciate the reviewer’s comments. Following his suggestion, we added a brief description of the training program, as presented in the cited references.

Reviewer 2 Report

Please find the information below as an attempt to help refine the product that has been offered. 

Abstract –

Everything seems appropriate

Introduction –

Lines 48-51 – need citations

Line 56 – avoid ‘their, they, them’ these words are vague and detract from the pointed information meant to be conveyed.

Line 62 – I am not sure if Soldiers needs to be capitalized.

Line 72 – random 26

Line 96/97 – Does there really need to be a break in the paragraph here?

Methods -

Line 162 – Is Brazilian relevant to this point?

Results –

Figure 1 is great!

Discussion –

Line 230 – delete ‘ones’

Line 235 – Delete ‘Brazilian’. The investigation examined a sample. The fact that the sample os derived from a population in Brazil should not be germane to the report.

Line 245/246 – no need to break the paragraph

Line 290 – edit grammar.

General comment –

The use of nationalized labels is not relevant to the research or the discussion. Unless the authors can definitively indicate through deeply cited research that there is a physiologically significant difference between Brazilian, American and or Italian members of our species these labels should be deleted and rephrased to the context of the manuscript.

I do believe the authors have offer

Author Response

(R. = Reviewer; A. = Authors)

Reviewer 2

R.: Please find the information below as an attempt to help refine the product that has been offered.

A.:. We appreciate the reviewer's detailed review, and below, we responded point by point to the comments.

R.: Abstract –

  • Everything seems appropriate

R.: Introduction –

  • Lines 48-51 – need citations
  • :. Citation inserted.
  • Line 56 – avoid ‘their, they, them’ these words are vague and detract from the pointed information meant to be conveyed.
  • :. We understand the reviewer's concern about the use of pronouns for these situations, and we have made adjustments throughout the text, when necessary.
  • Line 62 – I am not sure if Soldiers needs to be capitalized.
  • :. Letter "s" changed to lowercase.
  • Line 72 – random 26
  • :. number 26 excluded.
  • Line 96/97 – Does there really need to be a break in the paragraph here?
  • :. We agree with the reviewer. We changed the paragraphs, leaving them in only one.

R.: Methods -

  • Line 162 – Is Brazilian relevant to this point?
  • :. We agree with the reviewer. The word has been withdrawn.

R.: Results –

  • Figure 1 is great!
  • :. we are thankful for the complement.

R.: Discussion –

  • Line 230 – delete ‘ones’
  • :. Done.
  • Line 235 – Delete ‘Brazilian’. The investigation examined a sample. The fact that the sample os derived from a population in Brazil should not be germane to the report.
  • :. Done.
  • Line 245/246 – no need to break the paragraph
  • :. The paragraphs have been merged.
  • Line 290 – edit grammar.
  • :. We have changed the begin of the paragraph to “The main strengths of the present research are related to the application...”.

R.: General comment –

  • The use of nationalized labels is not relevant to the research or the discussion. Unless the authors can definitively indicate through deeply cited research that there is a physiologically significant difference between Brazilian, American and or Italian members of our species these labels should be deleted and rephrased to the context of the manuscript.
  • :. According to the reviewer's suggestions, we disregarded the origin of the sample subjects in the text.

Reviewer 3 Report

Dear Authors

You have written an interesting study. However, several parts need to be addressed for greater clarity and reproducibility.

Introduction:

The first paragraph needs referencing. Also, it is poorly designed. Where is the main rationale for this study? Report what body composition values have been previously reported. This is then the baseline for how the body composition is measured.  Amend accordingly

The sentence in line 58 needs referencing.

Line 72 - what is the meaning of the number 26? Elaborate and correct

Lines 73-78 need to be supported by references. Add

Overall the introduction is missing the information on how previous studies reported validity and reliability between your used methods, or are you the first one? Amed accordingly

Methods:

Military physical training description needs to be after the presentation of the sample. Additionally, it is not enough just to put references about one of the most important variables - training. The training needs to be better described. Amend accordingly

The sample - How did you calculate the sample size for your study (G*Power or any other method)? Report

Inclusion criteria need to be better defined - what about the presence or history of musculoskeletal injuries? report

SMT group - how many per particular sport? Report

On what ground were the participants divided into those 2 groups (randomly or any other reason – previous activity in selected sports)? Report

Measurements:

  • What was the sequence of measurements - anthropometry, DXA and BIA? Clearly state that - or did you have a random order? Report
  • What was the break between measurements? report
  • There is nowhere reported what were the instructions to the participants one day before the measurements - level of activity, nutrition, what were the instructions before the measurements - restriction of activity, alcohol consumption, emptying of bowel and bladder 30 min before measurements, etc (these conditions highly affect body composition measurements in your techniques). Were the manufacturer instruction followed for accurate measurements - from your description this is not visible. Amend accordingly in every test setting
  • report the model of weight scale and stadiometer
  • (BMI, kg·m-2) the abbreviation is not at the right place  - put it directly after you mention body mass index

  • who performed the measurements (were they ISAK accredited)? report
  •  ''Relaxed upper arm, waist and calf girths'' were measured in which places? according to ISAK? How many times, what value was taken into further analysis? Why just these parameters? Report and be precise

  • At what time of the day was this done? report
  • DXA reproducibility was done how? Again be precise - how long apart were these two measurements on those 23 subjects? Report
  • Report validity of your device and not just reliability
  • BIA measurements - in which position were participants tested, instructions 30min before testing. Report validity and reliability of your device.

Statistical analysis

The use of Correlations is not enough to compare or check the measurements of the two methods - validity. This does not check if the methods had a high agreement. Additional tests need to be performed as paired samples T-test, Crombach alpha, coefficient of determination (R2) and Bland Altman plots. Amend your methods and results accordingly (Table 2).

Results

Table 1 - report 95% confidence interval for the variables

Upper arm and calf girth are missing as variables you mentioned in methods! Add

Phase needs to be explained in the legend or add angle - or use correct abbreviations PA as you use in other tables

Table 2 reports results at baseline - what about the results after the 7 months of training? This is missing add data

Table 3 - Upper arm and calf girth are missing as variables you mentioned in methods! Add

Discussion

First paragraph - You can't conclude that DXA and BIA had high agreement just on the basis of correlation - perform additional analysis.

Lines 242-245 need referencing. Add

Limitations of the study need to extend with the lack of guidelines before the tests were performed as this drastically affects the study's results.

Overall the discussion is poorly written and does not connect the variables and training. Also, it does not address specific training of the second group where some athletes specialised from combat sports to endurance disciplines and how this could affect results.

Additionally no comment on the training itself and what could be changed to achieve better results. This component is completely missing.

Line 301 - military people? use population

Overall a poorly written paper. However, I see potential and I want to give authors a chance to address raised comments and questions adequately.

Therefore, I recommend reconsideration after a major revision

Author Response

(R. = Reviewer; A. = Authors)

Reviewer 3

R.: Dear Authors, You have written an interesting study. However, several parts need to be addressed for greater clarity and reproducibility.

A.:. We appreciate the reviewer's detailed review, and below, we responded point by point to the comments.

R.: Introduction:

  • The first paragraph needs referencing. Also, it is poorly designed. Where is the main rationale for this study? Report what body composition values have been previously reported. This is then the baseline for how the body composition is measured. Amend accordingly
  • : We have added the citation. However, it is not the purpose of the first paragraph to present the rationale of the study, nor to identify all the variables of body composition to be used. We understand that this first paragraph was written especially for the reader to position himself in front of the rest of the text, mainly in the Introduction, which, yes, contains the information that the reviewer questioned.
  • The sentence in line 58 needs referencing.
  • : The references are in line 59 – [6-8].
  • Line 72 - what is the meaning of the number 26? Elaborate and correct
  • : We have withdrawn the number; it was a typo.
  • Lines 73-78 need to be supported by references. Add
  • : As suggested by the reviewer, we have added a reference.
  • Overall the introduction is missing the information on how previous studies reported validity and reliability between your used methods, or are you the first one? Amed accordingly
  • : The information is in lines 93-98.

R.: Methods:

  • Military physical training description needs to be after the presentation of the sample. Additionally, it is not enough just to put references about one of the most important variables - training. The training needs to be better described. Amend accordingly
  • : As suggested by the reviewer, we have added a brief description of the military training and rearranged the text position on the section.
  • The sample - How did you calculate the sample size for your study (G*Power or any other method)? Report
  • : As presented in the text, the subjects belong to a convenience sample (all admitted cadets were recruited for evaluation, except for those who met the exclusion criteria). There was no sample size calculation.
  • Inclusion criteria need to be better defined - what about the presence or history of musculoskeletal injuries?
  • : As mentioned in the previous item, the inclusion criterion was to be admitted to the military school. We added the question related to the history of musculoskeletal injury in the exclusion criterion, since this criterion was considered for the evaluation of the cadets.
  • SMT group - how many per particular sport?
  • : We have added the information.
  • On what ground were the participants divided into those 2 groups (randomly or any other reason – previous activity in selected sports)?
  • : Participants, upon admission to the military college, choose or not for the group of sports activities, and for the specific sport. So, there is no randomization. So that this information could be understood, we inserted the phrase " by their own choice" in the description of group 2. Thus, the new text was as follows: "The sample was divided into two groups: 1) the cadets who were involved in the military physical training routine only (MT, n = 155); 2) the cadets who were involved, by their own choice, in the military physical training routine plus a specific sport training for military competition".
  • What was the sequence of measurements - anthropometry, DXA and BIA? Clearly state that - or did you have a random order? Report what was the break between measurements?
  • : We have inserted the information in the text.
  • There is nowhere reported what were the instructions to the participants one day before the measurements - level of activity, nutrition, what were the instructions before the measurements - restriction of activity, alcohol consumption, emptying of bowel and bladder 30 min before measurements, etc (these conditions highly affect body composition measurements in your techniques). Were the manufacturer instruction followed for accurate measurements - from your description this is not visible. Amend accordingly in every test setting
  • : All instructions suggested were included.
  • report the model of weight scale and stadiometer
  • : Models reported.
  • (BMI, kg·m-2) the abbreviation is not at the right place - put it directly after you mention body mass index
  • : Done.
  • who performed the measurements (were they ISAK accredited)? report
  • : Yes, all the technicians of the laboratory were accredited by the ISAK by a national accredited examiner.
  • ''Relaxed upper arm, waist and calf girths'' were measured in which places? according to ISAK? How many times, what value was taken into further analysis? Why just these parameters? Report and be precise
  • : The main variable of the study is the Specific bioelectrical impedance vector analysis, when Rsp and Xcsp values are presented; for the calculation of these variables, according to the proposal of the authors who proposed it, the measurement of relaxed arm, waist and calf circumference are necessary, according to the proposal of Lohman (1988), for the adjustment of the variables for each cylinder of the body (according to the authors' biophysical assumptions), as can be verified in the text:
    • “Specific bioelectrical impedance vector analysis was applied [23]. Specific bioelectrical values (resistivity [Rsp] ohm cm; reactivity [Xcsp] ohm cm) were obtained by multiplying resistance and reactance by a correction factor (A/L), where area (A, cm2) and length (L, cm) were estimated as follows: A = (0.45 upper arm area + 0.10 waist area + 0.45 calf area) and L=1.1 stature (in cm). The segment areas were calculated as C2/4π, where C (cm) is the girth of the upper arm, waist, or calf.”
  • Also, for this reason, the circumference variables were not presented in the tables (according to the reviewer's suggestion), as they are only used for adjustment, and not for characterization. Still answering the reviewer's questions, they were performed in triplicate and rotationally, and the median value of the three measurements was considered.
  • At what time of the day was this done? report
  • : All measurements were performed in the morning, as presented in the first paragraph of the Measurements section.
  • DXA reproducibility was done how? Again be precise - how long apart were these two measurements on those 23 subjects? Report
  • : The 23 evaluated were reassessed within 24 hours; information inserted in the text.
  • Report validity of your device and not just reliability
  • : In order to have the validity of our equipment, we need to have a method considered "gold standard" for comparison, which was not the case; in this sense, we do not have the information.
  • BIA measurements - in which position were participants tested, instructions 30min before testing. Report validity and reliability of your device.
  • : Information about BIA measurements and test reliability ere included in the test.
  • Statistical analysis: The use of Correlations is not enough to compare or check the measurements of the two methods - validity. This does not check if the methods had a high agreement. Additional tests need to be performed as paired samples T-test, Crombach alpha, coefficient of determination (R2) and Bland Altman plots. Amend your methods and results accordingly (Table 2).
  • : This is not a validation study. Information regarding the correlation coefficient was presented as complementary information. The BIA variables (Rsp and Xcsp) are raw impedance values, in ohms. The DXA variables are body composition values, measured in kg. Statistically, it is possible to perform the T-test paired samples tests and Bland-Altman, however, it is not plausible, as these are variables with different outcomes, therefore, the results will not present any interesting analysis for the reader. For example, the Bland-Altman is the graph plotting the values of the difference between the two methods on the Y axis, and the average of the two methods on the X axis. It makes no sense to calculate the difference between ohms and kg, nor does it make sense to calculate the average. The same can be applied for the comparison test, when there will certainly be statistically significant differences and to the Cronbach Alpha Coefficient.

R.: Results

  • Table 1 - report 95% confidence interval for the variables
  • : Done.
  • Upper arm and calf girth are missing as variables you mentioned in methods! Add
  • : As previously presented, these variables are only for adjusting the Rsp and Xcsp variables, so we chose not to insert them in the text, so as not to inflate the table with excessive information.
  • Phase needs to be explained in the legend or add angle - or use correct abbreviations PA as you use in other tables
  • : Variable has been renamed.
  • Table 2 reports results at baseline - what about the results after the 7 months of training? This is missing add data
  • : Table 2 was made to confirm the proposal to use specific BIVA for, especially, body composition components, as suggested by Buffa et al. (2013). Therefore, it was made only for the baseline, to confirm its correlation.
  • Table 3 - Upper arm and calf girth are missing as variables you mentioned in methods! Add
  • : As previously presented, these variables are only for adjusting the Rsp and Xcsp variables, so we chose not to insert them in the text, so as not to inflate the table with excessive information.

R.: Discussion

  • First paragraph - You can't conclude that DXA and BIA had high agreement just on the basis of correlation - perform additional analysis.
  • : As previously described, the suggested analyzes cannot be performed in the present study, because the variables have different outcomes. At the reviewer's suggestion, we changed the term agreement to correlated or correlation, according to the meaning of the phrase.
  • Lines 242-245 need referencing. Add
  • : References added.
  • Limitations of the study need to extend with the lack of guidelines before the tests were performed as this drastically affects the study's results.
  • : Once the suggestions of the reviewer about these guidelines were accepted and included in the text, it will be not necessary to extend the limitations of the study.
  • Overall the discussion is poorly written and does not connect the variables and training. Also, it does not address specific training of the second group where some athletes specialized from combat sports to endurance disciplines and how this could affect results. Additionally no comment on the training itself and what could be changed to achieve better results. This component is completely missing.
  • : We respect the reviewer's opinion on the quality of the discussion. Considering the characteristics of the journal (IJERPH), we adopted the discussion much more focused on variations in body composition and comparison between evaluation methods, than the descriptions of the types of training itself, especially since the sports involved were different. many, and distinct, when considering mechanical actions and physiological demands, which can influence responses on body composition, and would, consequently, make the discussion extremely large and outside the acceptable limits by the IJERPH. However, even so, we changed some parts of the text, and inserted, under the reviewer's recommendation, new references.
  • Line 301 - military people? use population
  • : Done.

Overall a poorly written paper. However, I see potential and I want to give authors a chance to address raised comments and questions adequately. Therefore, I recommend reconsideration after a major revision.

A.:. We fully respect the reviewer's opinion. Within what has been suggested to us and what we consider appropriate for the article, we accept and are extremely grateful.

Round 2

Reviewer 3 Report

Dear Authors,

Thank you for addressing the majority of my questions and suggestions.

However, some parts have still not been addressed adequately.

  • There is nowhere reported what were the instructions to the participants one day before the measurements - level of activity, nutrition, what were the instructions before the measurements - restriction of activity, alcohol consumption, emptying of bowel and bladder 30 min before measurements, etc (these conditions highly affect body composition measurements in your techniques). Were the manufacturer instruction followed for accurate measurements - from your description this is not visible. Amend accordingly in every test setting
  • : All instructions suggested were included.

The included paragraph needs referencing. Add

  • Report validity of your device and not just reliability
  • : In order to have the validity of our equipment, we need to have a method considered "gold standard" for comparison, which was not the case; in this sense, we do not have the information

You can use the validity studies of the same equipment done by other studies. Report validity.

  • statistical analysis: The use of Correlations is not enough to compare or check the measurements of the two methods - validity. This does not check if the methods had a high agreement. Additional tests need to be performed as paired samples T-test, Crombach alpha, coefficient of determination (R2) and Bland Altman plots. Amend your methods and results accordingly (Table 2).
  • : This is not a validation study. Information regarding the correlation coefficient was presented as complementary information. The BIA variables (Rsp and Xcsp) are raw impedance values, in ohms. The DXA variables are body composition values, measured in kg. Statistically, it is possible to perform the T-test paired samples tests and Bland-Altman, however, it is not plausible, as these are variables with different outcomes, therefore, the results will not present any interesting analysis for the reader. For example, the Bland-Altman is the graph plotting the values of the difference between the two methods on the Y axis, and the average of the two methods on the X axis. It makes no sense to calculate the difference between ohms and kg, nor does it make sense to calculate the average. The same can be applied for the comparison test, when there will certainly be statistically significant differences and to the Cronbach Alpha Coefficient.

I can't agree with your answer. Your first aim in the study is: to analyze the suitability of spBIVA to detect changes in body composition compared to DXA in a Brazilian military sample. What is that if not validity? Correlation does not imply causation and can't be the only tool to check two technologies. Therefore, amending the statistical section and results accordingly to my first questions.

I still think the Overall discussion is poorly written and does not connect the variables and training as the main variable even with those few added sentences. Try to extend it with the increased number of physical activities in military personnel and the effect on body composition.

Overall the paper is heading in the right direction, however still needs further work. Therefore I recommend major revision.

Kind regards

Author Response

(R2. = Reviewer Round 2; A2. = Authors Round 2)

Reviewer 3

R2.: Thank you for addressing the majority of my questions and suggestions. However, some parts have still not been addressed adequately.

A2.: The authors thank the reviewer for the recommendations for improving the article.

R2.: The included paragraph needs referencing. Add

A2.: Reference included – Kyle et al., 2004 [37].

R2.: You can use the validity studies of the same equipment done by other studies. Report validity.

A2.: Validity reported – Gonzalez et al., 2019 [38].

 R2.: I can't agree with your answer. Your first aim in the study is: to analyze the suitability of spBIVA to detect changes in body composition compared to DXA in a Brazilian military sample. What is that if not validity? Correlation does not imply causation and can't be the only tool to check two technologies. Therefore, amending the statistical section and results accordingly to my first questions.

A2.: We understand that your concern is with the aim of the study being related to "validation". In this sense, we changed the objective to correlation, since the proposed statistic definitely cannot be done, as we have already explained in the previous review.

A previous study was performed to actually validate (BUFFA et al., 2013), where - using receiver operating characteristic (ROC) curves - specific BIVA showed to be accurate (considering specificity and sensitivity) in evaluating FM% against DXA (ROC areas: 0.84–0.92). In the same study the authors have also evaluated the accuracy in evaluating ECW/ICW (ROC areas between 0.83 and 0.96) and showed that the accuracy was similar in the two sexes (p = 0.144). It is noteworthy to mention that two authors of this study are in the current study.

R2.: I still think the Overall discussion is poorly written and does not connect the variables and training as the main variable even with those few added sentences. Try to extend it with the increased number of physical activities in military personnel and the effect on body composition.

A2.: The discussion has been improved accordingly.

R2.: Overall the paper is heading in the right direction, however still needs further work. Therefore I recommend major revision. Kind regards

A2.: Once again, we are very grateful to the reviewer for his attention to our article. It certainly helped to improve its quality.

This manuscript is a resubmission of an earlier submission. The following is a list of the peer review reports and author responses from that submission.

Round 1

Reviewer 1 Report

This reviewer wants to congratulate the authors on this research. I have detected some important limitations both in the description of the training program and in the selection of the sample. I think this must be clarified so that it can be published in a scientific journal.

Line 58 to 67 approx seems more a part of the discussion than a part of the introduction. I suggest removing from introduction and integrating this paragraph in the discussion.

Can you provide the number of the ethics code? “The research was approved by the Ethics Committee of the 106 School of Medical Sciences, University of Campinas. With number ¿?

Line 110. Military Physical Training

This is one of the core parts of the study, so authors should indicate more specifically the characteristics of training, which calisthenic exercises? what muscle groups? Is the program structured or do they always do the same? Indoor/outdoor/mix?? the training manual to which they refer, is it published? can you provide the bibliographic reference? During training, the military has been monitored in some way (HR, RPE ???) if not, it should include it in limitations.

Line 117. Sample. It seems that the data comes from 7 and 8 years ago. This seems like a limitation to me. Do you believe that things could have changed something, for example, the greater reliability of measuring instruments today? This should be discussed or added as limitation.

Line 123. Sample. I have a question with this group "the cadets who were involved in the military physical training routine plus a specific sports training for military competition" for example, I do not understand if the cadets who play chess should remain in the same group as the triathlon, or how in team sports, what is the difference between players who play all games and reservists or occasional players

Reviewer 2 Report

Define if it was 7 months or 1 year of training (page 3, rows 10 and 122) 

Remove rows 156-171 from the statistical analysis section

Interesting work, but with low originality. The fact that BIVA has not been used in military personnel is irrelevant, and more so because this work included young applicants from a preparatory school. It would be a very good proposal use spBIVA to evaluate and determine the impact of specialized tactical training on the health of active personnel belonging to the army, as well as for the annual or periodic medical evaluations to detect overweight or obese military personnel and overtraining states.

Reviewer 3 Report

The information presented is interesting. 

I suggest a thorough editing for grammar. There are several points that need addressing for text agreement and appropriate word choice. 

Additionally, there is information presented that requires citations/references. 

When these changes are made, perhaps the document will be more readable and thus convey the information in a smoother manner. In the current state the document is difficult to read through as it is an effort to truly understand the points being made. 

An expansion of the conclusion section is warranted.